# Reverse time migration (RTM) imaging of iron-oxide deposits in the Ludvika mining area, Sweden

Yinshuai Ding[1] and Alireza Malehmir[1]

1, Department of Earth Sciences, Uppsala University, SE 75236, Uppsala, Sweden

*Correspondence to*: Yinshuai Ding (yinshuai.ding@geo.uu.se)

**Abstract.** To discover or delineate mineral deposits and other geological features such as faults and lithological boundaries in their host rocks, seismic methods are preferred for imaging the targets at great depth.. One major goal for seismic methods is to produce a reliable image of the reflectors underground given the typical discontinuous geology in crystalline environment with low signal-to-noise ratio. In this study, we investigate the usefulness of reverse time migration (RTM) imaging algorithm in hardrock environment by applying it to a 2D dataset, which was acquired in the Ludvika mining area of central Sweden. We provide a how-to solution for applications of RTM in future and similar datasets. When using the RTM imaging technique properly, it is possible to obtain high-fidelity seismic images of the subsurface. Due to good amplitude preservation in the RTM image, the imaged reflectors provide indications to infer their geological origin. In order to obtain the reliable RTM image, we performed a detailed data pre-processing flow to deal with the random noises, near-surface effects and irregular receiver spacing and source spacing which are possible to downgrade the final image if ignored. Exemplified with the Ludvika data, the resultant RTM image not only delineates the iron-oxide deposits down to 1200 m depth as shown from previous studies, but also provides a better inferred ending of sheet-like mineralization. It also provides a much-improved image of the lithological contacts and crosscutting features relative to the mineralized sheets when compared to the images produced by Kirchhoff migration in the previous studies. Two of the imaged crosscutting features are considered to be crucial when interpreting large-scale geological structures in the site and the likely disappearance of mineralization at depth. Using a field data acquired in hardrock environment, we demonstrate the usefulness of RTM imaging workflows for deep targeting mineral deposits

## 1 Introduction

Seismic methods are favourable for deep targeting in mineral exploration, because of their ability to image targets at great depth (>500 m). (Malehmir et al., 2012b and references therein). Compared to other geophysical methods (e.g., gravity, magnetic and/or electromagnetic), the seismic methods investigate the properties (i.e, impedance) of the subsurface in a wave-

equation-based way during seismic data processing and imaging(Eaton et al., 2003). Generally speaking, due to the less attenuation effects of the seismic waves as a function of depth compared to EM methods, they tend to hold better resolution at depth than EM methods although they do have different sensitivity to different properties. Seismic methods may provide an image of the targets with high resolution at depth when the survey is designed to record the signal reflected from the targets directly below the survey area/line.. In such a seismic survey, a seismic source(e.g., a sudden impact produced by drophammer) should be employed to generate seismic wavefields with sufficient energy and frequency bandwidth (e.g., Brodic et al., 2019; Pertuz et al., 2020). The adequate energy ensures the wavefields propagate down to the depth of targets and travel back to the surface. A reasonable frequency bandwidth of the wavefields contributes to a suitable resolution image of the target area (ten Kroode et al., 2013).

Although the seismic methods have been well established and proven successful to delineate complex geological structures in sedimentary basins for hydrocarbon exploration (e.g., Sheriff and Geldart, 1995), their applications to delineate deep targets for mineral exploration are still relatively limited (Malehmir et al., 2012a and references therein; Buske et al., 2015; Malehmir et al., 2020a and references therein). There are several reasons for this hurdle. First, exploring deep-seated deposits is economically costlier compared to that at the shallow subsurface (<500 m). Second, the strong scattered waves due to abundant small heterogeneities and complex structures in hardrock environment (Cheraghi et al., 2013; Bellefleur et al., 2018; Bräunig et al., 2020), will contaminate the reflections from targets of mineralization and cause difficulties to image the targets. However, exploration of mineral deposits (especially the deep ones) using seismic methods has become more appreciated by both the industry and the academia (Malehmir et al., 2020 and references therein). Two main factors can be accounted for this success. First, seismic data acquisition and processing have become much cheaper and more affordable by exploration companies. Still, mining companies are to be encouraged to employ the seismic method more routinely. Second, seismic imaging techniques have computationally been feasible and well developed to handle complex subsurface structures (O'Brien, 1983; Bednar, 2005). Regardless, the potential of prestack depth imaging algorithms (e.g., Kirchhoff and RTM) in imaging hardrock environment still need to be explored with effort and case studies. Though Kirchhoff prestack depth imaging algorithms have been attempted with a few good illustrating results (Bellefleur et al., 2018; Bräunig et al., 2020), direct targeting deep mineral deposits using the RTM methods has rarely been applied to mineral exploration data examples.

In this study, our main goal is to demonstrate the usefulness of the seismic methods in deep-targeting of iron-oxide deposits with the employment of the RTM imaging algorithm (Baysal et al., 1983; Zhou et al., 2018) in a field dataset. The dataset was acquired in the Ludvika mining area (Blötberget), in the central Sweden in 2016. Although there have been several studies (Balestrini et al., 2020; Bräunig et al., 2020; Markovic et al., 2020; Maries et al., 2020) on this 2D dataset, RTM as a method honouring the two-way wave equation has not been applied to it yet. In this work, we show that advanced imaging methods such as RTM can produce qualified images to aid better understanding of the deposits and other geological features in the host rocks.

This paper covers three key parts. First, we provide a brief review of the study area. Second, we apply the RTM imaging method to the dataset after a comprehensive data pre-processing. Third, we study and interpret the resultant RTM image by integrating with other geological and geophysical datasets.

## 2 Study area

The Ludvika mining area belongs to the Paleoproterozoic Bergslagen mineral district (Figure 1), which has a long history for iron ore mining (Magnusson, 1970). The ore deposits are primarily magnetite and hematite as sheet-like bodies. At the Blötberget site where this study focuses, the sheet-like mineral horizons show a moderate dip (around $30 - 45^o$) towards the south-southwest based on the core logging data (Maries et al., 2017) and a recent 3D reflection survey in the area ( Malehmir et al., 2020b).

Due to the dropping steel prices on the global market, the mining operations had to be stopped in 1979. However, because of recent feasibility studies and more available resources, as well as the growing iron ore market, Nordic Iron Ore (NIO) plans to reopen the mine and exploit the deposits at depth level of 400-420 m in the near future. Relevant to this study is the work by Maries et al. (2017) on downhole logging study of the iron-oxide deposits at the site. As shown in the logging data, the strong density contrast between the iron-oxide deposits ($\sim 4500$ kg m$^{-3}$) and their igneous host rocks ($\sim 2500$ kg m$^{-3}$) should produce a strong impedance contrast as a good target for the seismic methods. However, the resistivity of the deposits is not considerably strong as normally targeted for base metals (it is only on the order of 1000 ohm.m) and very similar to that of the host rocks (around 3000-5000 ohm per meter) to make the resistivity-based methods (such as EM) suitable for depth delineation of deposits.

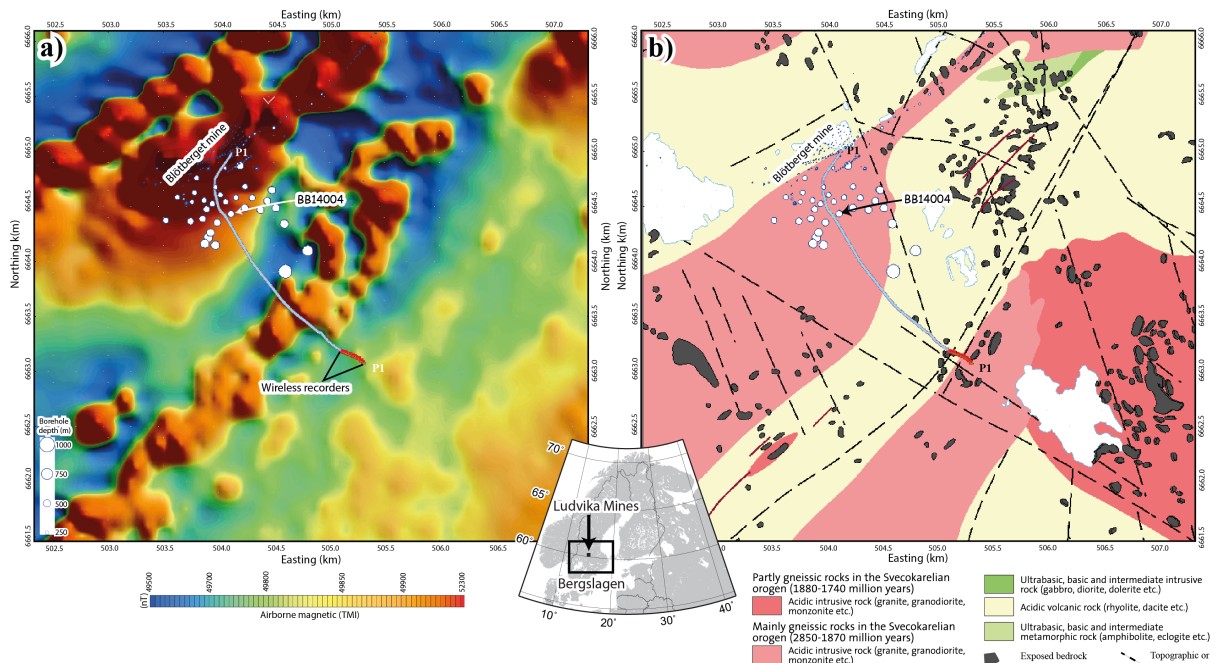

**Figure 1.** (a) Total-field aeromagnetic and (b) geological maps of Blötberget within the Ludvika Mines of Bergslagen mineral district in
central Sweden. 2D seismic survey (blue line) and borehole BB14004 are used in this study. Magnetic data were provided by the
Geological Survey of Sweden.
**3 RTM imaging on the legacy Blötberget dataset**
**3.1 Review of the seismic data acquisition**
The seismic survey was conducted along a 2D profile in 2016 (e.g., Markovic et al., 2020) following a feasibility study of a
seismic landstreamer survey (Malehmir et al., 2017). In order to better image the deposits, the seismic profile was designed to
run approximately perpendicular to the strike of the known mineralized sheets. In the survey, 451 receivers were deployed
(Figure 1) of which 427 were cable-connected geophones (blue line) and 24 were wireless recorders (red line) deployed at the
southern end of the profile. A 10-Hz spike-type geophone was used as the sensor. As for the receiver spacing, the cabled
recorders were deployed at every 5 m approximately while the wireless recorders at every 10 m approximately. The source
used was a 500-kg Bobcat-mounted drophammer. The source locations were collocated with the receiver locations. The
sampling rate was set as 1 ms and recording time to 2 s. To avoid violating the 2D assumption of the subsurface below the
seismic profile, we only use 369 receivers and sources that form a rather straight line in this study. The remaining 82 receivers
and sources are not used due to the fact that they make a strong bend towards the northern-west end of the profile. Details of
the survey can be found in Markovic et al. (2020) and Maries et al. (2020).

## 3.2 Data Pre-processing

The aim of data pre-processing is to strengthen the reflected events in terms of amplitudes and continuity. For the pre-processing of the seismic data, we designed a 6-step workflow as shown in Figure 2. Though we chose the specific processing method for every step, it is possible to use different methods for pre-processing when done carefully.

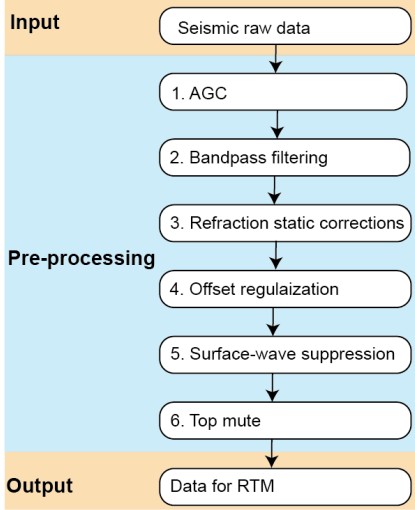

**Figure 2.** The six-step data pre-processing workflow for the RTM imaging. In this study, offset regularization and surface-wave suppression had the most significant roles.

**Step 1.** We applied automatic gain control (AGC) (e.g., Yilmaz, 2001) to balance the amplitudes at different offsets and different arrival times. The amplitudes of the raw data (Figure 3a) are more balanced after AGC (Figure 3b), though noise is also amplified at the same time. This step was necessary to reduce the amplitude of the surface-waves as they dominate the seismic signal.

**Step 2.** We used a bandpass filter (10-35-150-200 Hz) to maintain data in this frequency band based on the analysis of the frequency band in the first breaks (Figure 3c).

**Step 3.** We performed surface-consistent refraction static corrections to compensate for the near surface effects due to the different surface topography and the low-velocity glacial covers. The irregularity in time of the first arrivals is mitigated after the refraction static corrections (Figure 3d).

**Step 4.** We practiced offset regularization of the dataset along a 1D smooth-curved line and obtain a regularized dataset (Figure 3e). The original locations of sources and receivers (Figure 4a) were projected onto the 1D curve which was defined by a

polynomial of degree 3 (Figure 4b). Based on the projected receiver locations (Figure 4c), we regularized the receiver spacing
to a constant interval (i.e., 5 m) along the 1D curve (Figure 4d). The seismic traces at those regularized receiver locations were
obtained by a cubic interpolation using the traces at the projected receiver locations. Similarly, we regularized the source
locations along the 1D curve and obtained the seismic traces at every 5m by a cubic interpolation which was performed in the
common receiver domain. After data regularization, we acquired a more even fold coverage required for RTM imaging. AS a
result, the total number of shot points (or receiver points) were regularized to 415 from the original 369. Such even fold
coverage should contribute to an amplitude-balanced image, although one needs to use and inspect the interpolated seismic
traces with care.
**Step 5.** We suppressed the surface-waves (Figure 3f) using the curvelet filtering method (Candès et al., 2006). The geometrical
difference of surface waves (almost linear events) and reflections (non-linear events) provides a feasibility to sperate them
using curvelet filtering. After removing the strong surface-waves with large amplitudes, the relative weaker reflection events
show up clearly in the dataset.
**Step 6.** We muted the direct arrivals and the ambient noise above the first breaks (Figure 3g) to reduce artefacts in the final
seismic section.

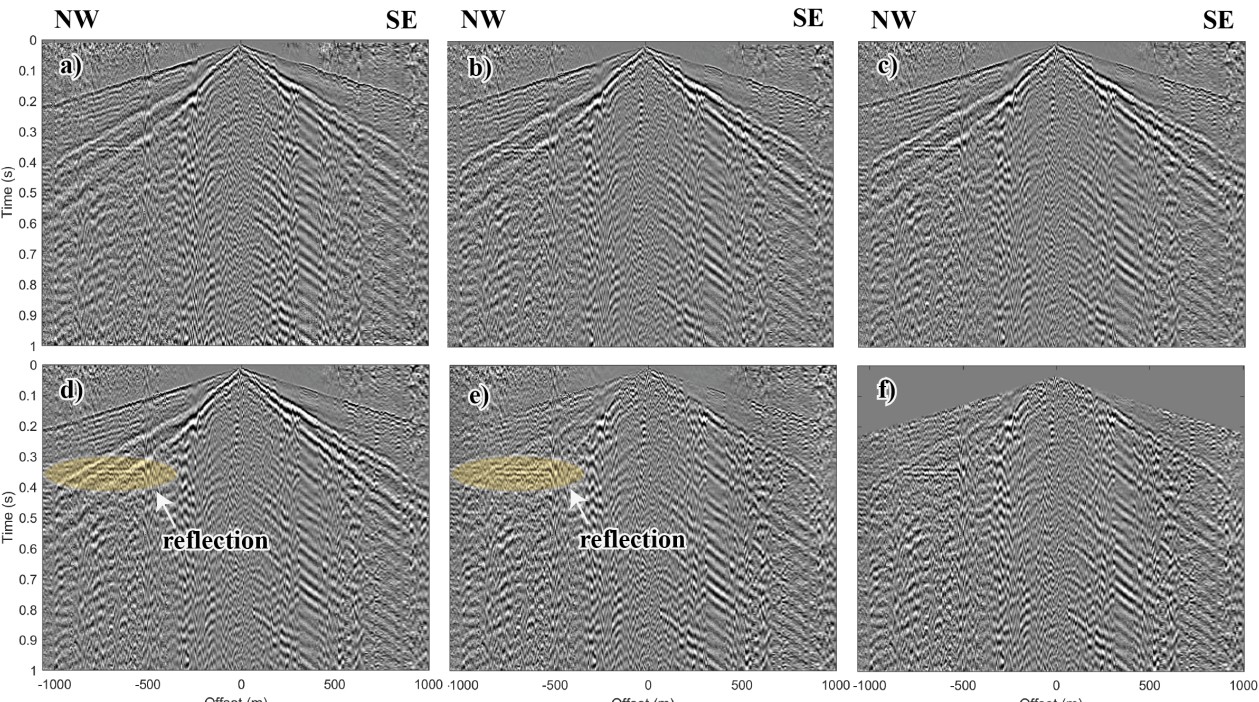

**Figure 3.** (a) Example raw shot gather after AGC application (Step 1). (b) After bandpass filtering (Step 2). (c) After refraction static
corrections (Step 3). (d) After data regularization (Step 4). (e) After surface-wave suppression (Step 5) and (f) after top muting (Step 6).
The yellow ellipses mark a clear reflection which we interpret to be from the iron-oxide deposits.

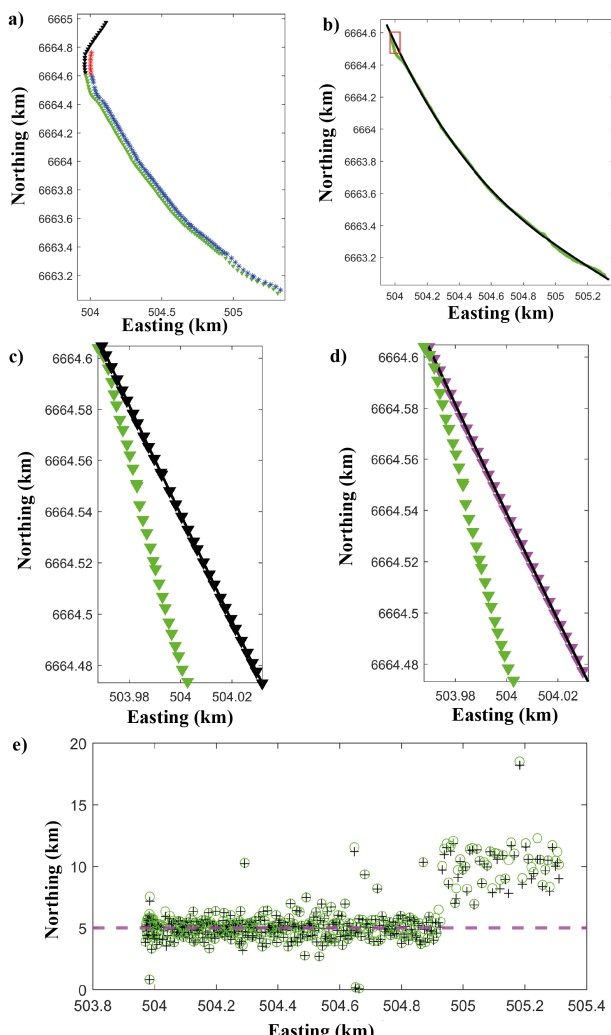

**Figure 4.** a) Original source (blue and red stars) and receiver (green and black triangles) locations. The blue sources and the green receivers were selected for RTM imaging. b) The 1D smooth-curved line (black) based on a line fitting method using the positions of green receivers. c) The receivers (black triangles) projected along the 1D curve from their original locations (green triangles), zoomed in the red triangle in Figure 4b. d) The receivers (magenta triangles) regularized along the 1D curve from the projected receivers (black triangles) in Figure 4c. e) The spacing of receivers in their original locations (green circles), their projected locations (black cross), and regularized locations (magenta dashed line).

Note that we only used a bandpass filter to supress parts of the noise. When forming an RTM image using the filed data in the next section, the random noise in the data tends to be cancelled because of the cross-correlation imaging condition. Though we chose the specific processing methods in the pre-processing steps to prepare our data for RTM, it is possible to use different methods for pre-processing when done carefully.

## 3.3 RTM imaging

RTM imaging algorithm requires three ingredients (1) a source wavelet, (2) a shot gather, and (3) a smooth but reasonable velocity model. Using the source wavelet and the migration velocity model, one can then forward propagate wavefields from the source side. Using the shot gather and the migration velocity model, the backward-propagating wavefields from the receiver side can then be modelled. Cross-correlation between the forward-propagating wavefields and the backward-propagating wavefields forms a partial RTM image from a single shot gather. During the cross-correlation, an illumination compensation using source-side wavefields is done to balance the amplitudes in the seismic section (e.g., Valenciano and Biondi, 2003). Such a deconvolution imaging condition (Claerbout, 1971) removes the source wavelet approximately and hence improves the resolution of the image. The final RTM section is obtained by summing all the partial RTM images of all the shot gathers.

For the Blötberget seismic dataset, we used a staggered finite-difference modelling method (2nd order in time, 4th order in space) to realize the RTM. The source signal was chosen to be a Ricker wavelet (Figure 5a) with a peak frequency of 70 Hz and with a time step of $dt = 5 * 10^{-4}$ s. The $dt$ used in RTM is half of the sampling rate in the data. Hence, our data was up-sampled to $dt$ by a linear interpolation before it was being back propagated. The size of the 2D migration velocity (Figure 5b) was set 401 (vertical) by 415 (horizontal). The grid interval was set 5 $m$ along both directions ($dx = dz = 5$ m). To obtain a good migration velocity model using this dataset, we performed the semblance velocity analysis (e.g., Zhou, 2014). In doing this, the reflected signal from the deposits was the main constraint. Thus, the velocity above the deposits were well constrained while the other areas are less. To mitigate the numerical artefacts from the boundaries of the velocity model, we added the perfectly matched layers (PML) (e.g., Komatitsch and Martin, 2007) on the four  edges of the velocity model. Each PML has a thickness of 50 grid points. With the 4-sides PML, the wavefield modelling had no free-surface condition.

We ran the RTM for all the 415 shot gathers to obtain 415 partial images. Then, we applied a Gaussian smoothing filter to suppress noise with smaller wavelengths ($\lambda < \sim 10$ m, i.e., 2-grid size) in the partial images. Summing the smoothed 415 partial images, we obtained the final RTM image (Figure 6). Using the dominant frequency (60 Hz) in the field data and the migration velocity (~6000 m/s) at depth, we estimated that the vertical resolution based on major wavelength as 100 m approximately.

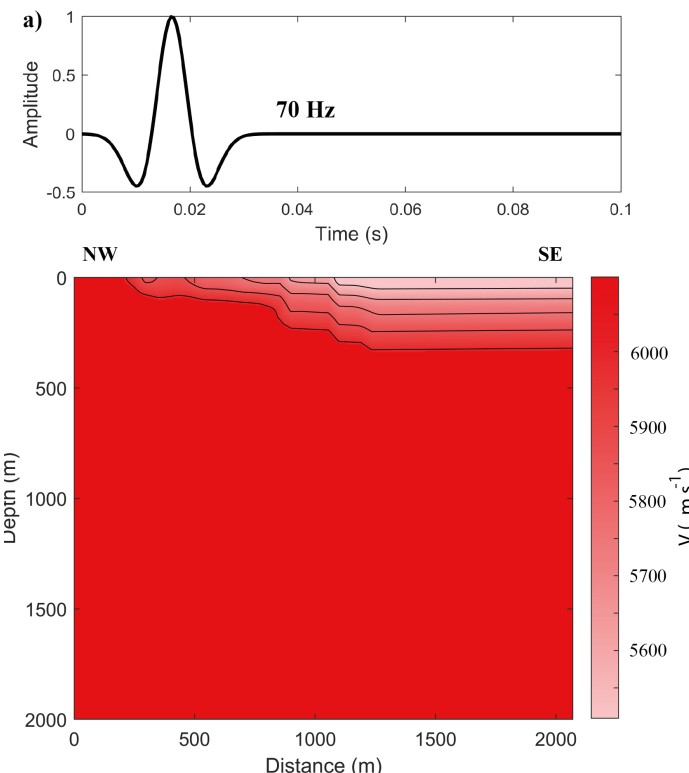

**Figure 5** (a) Ricker wavelet (70 Hz) was chosen as the source wavelet, which generates forward propagating wavefields. (b) Migration
velocity model used for the forward and backward propagations. The black lines are contour lines of the velocity values.

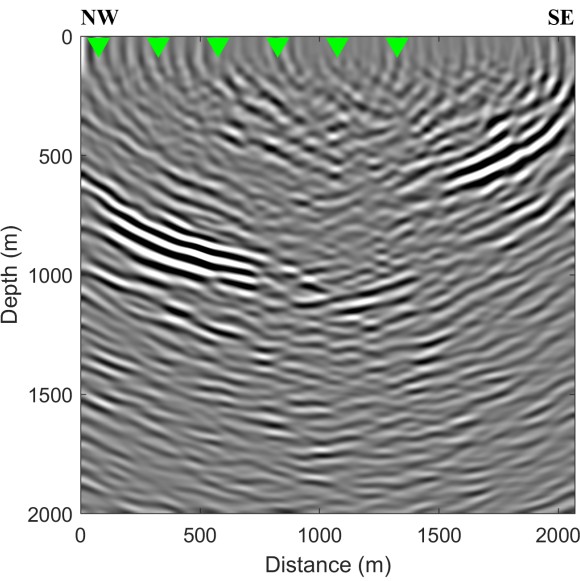

**Figure 6.** Final RTM section of the Blötberget legacy dataset. The green triangles are where common image gathers (CIG) are shown later
in the article.

## 4 RTM results, their interpretations and comparisons

Compared to the previous studies (Balestrini et al., 2020; Bräunig et al., 2020; Maries et al., 2020; Markovic et al., 2020) using the same dataset, the RTM image (Figure 6) is much more promising in terms of imaging the crosscutting features and rock contacts. We validated the trustworthiness of the imaged reflectors in the RTM image in two ways. First, we analysed six common image gathers (CIG) (e.g., Yan and Xie, 2011) from the image. Second, we integrated the results with other available geological and geophysical data to verify our interpretation of the reflectors in the section.

For a trace in the final image (Figure 6) at a specific spatial location (75, 325, 575, 825, 1075, and 1325 m distances along the profile), its corresponding common image gather (CIG) was formed by extracting the seismic traces from different partial images at the same spatial location. In our case, the CIG was composed of seismic traces extracted at those specific image locations from 415 single-shot partial images. In a single CIG gather, the traces were indexed by the offset between a source location and its specific CIG location in the RTM image. The flatness of continuous events in the CIG provided a qualitative evaluation of the fidelity of imaged reflectors. We extracted six CIGs from the RTM image (Figure 6) at 75, 325, 575, 825, 1075, and 1325 m (Figure 7). In the CIGs, the events reflected from the iron-oxide deposits are quite continuous and flat (Figure 7a-d) from zero offset to the far offset (~ 900 m), while events reflected from the inferred faults (Markovic et al., 2020) do not show continuity across the whole section (Figure 7e & 6f). The less continuous flattened events in the CIG corresponding to the inferred fault planes are likely due to the inaccuracy of the velocity model in the area of the inferred faults.

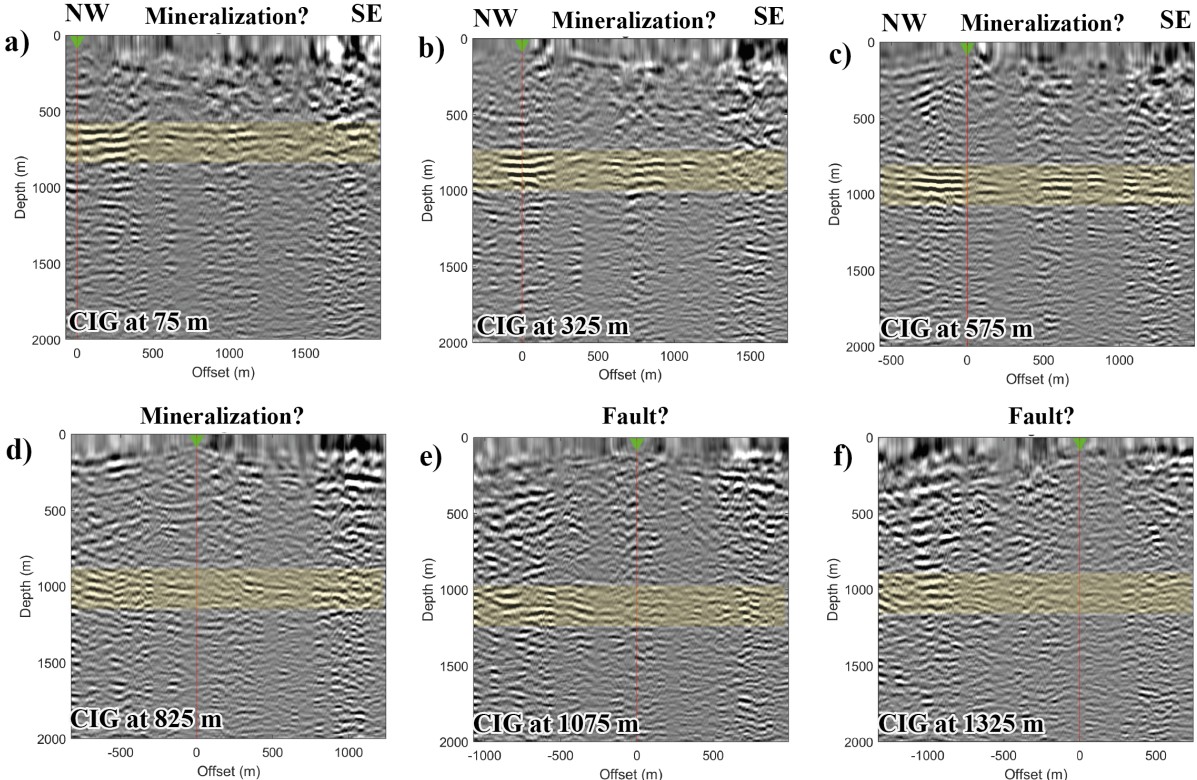

**Figure 7.** Six CIG gathers (a) 75 m, (b) 325 m, (c) 575 m, (d) 825 m, (e) 1075 m and (f) 1325 m distances along the RTM section extracted to present the quality of the RTM section (see Figure 5). The green receiver in each CIG collocates with their CIG positions in the original image. The yellow boxes highlight where the flat events are present.

The CIGs only show the trustworthiness of the seismic image in an image-wise sense. Only from the seismic image itself, it is difficult to deduce what the imaged reflectors represent geologically. Hence, one should also integrate the results with other geological and/or geophysical data to assist and validate the interpretations of the seismic image. In our study area, four other datasets are available for this purpose. First, we have an existing 3D ore block model from the boreholes. Second, two inferred 3D fault planes have recently been imaged and picked from the recent 3D survey (Malehmir et al., 2020b). Third, one borehole (BB-14004) near the seismic acquisition line has the natural gamma, core logging, density and sonic logging data (Figure 8) (Maries et al., 2017). Using the density data and P-wave sonic data, we calculated their reflection coefficients (Figure 8d) along the wellbore trajectories. Convolving the calculated reflection coefficients (Figure 8d) with a 70-Hz Ricker wavelet, we obtained a synthetic seismic trace along the wellbore trajectory (Figure 8e). Fourth, there is a high-resolution (200 m flight spacing) aeromagnetic data (Figure 9) available in this study area. The calculated magnetic anomaly data from it are used as well for the interpretation. Treating the RTM image as a dataset, we have 5 datasets in total. Assembling these 5 different datasets in 3D (Figure 9), we can note their spatial relationships and correlations.

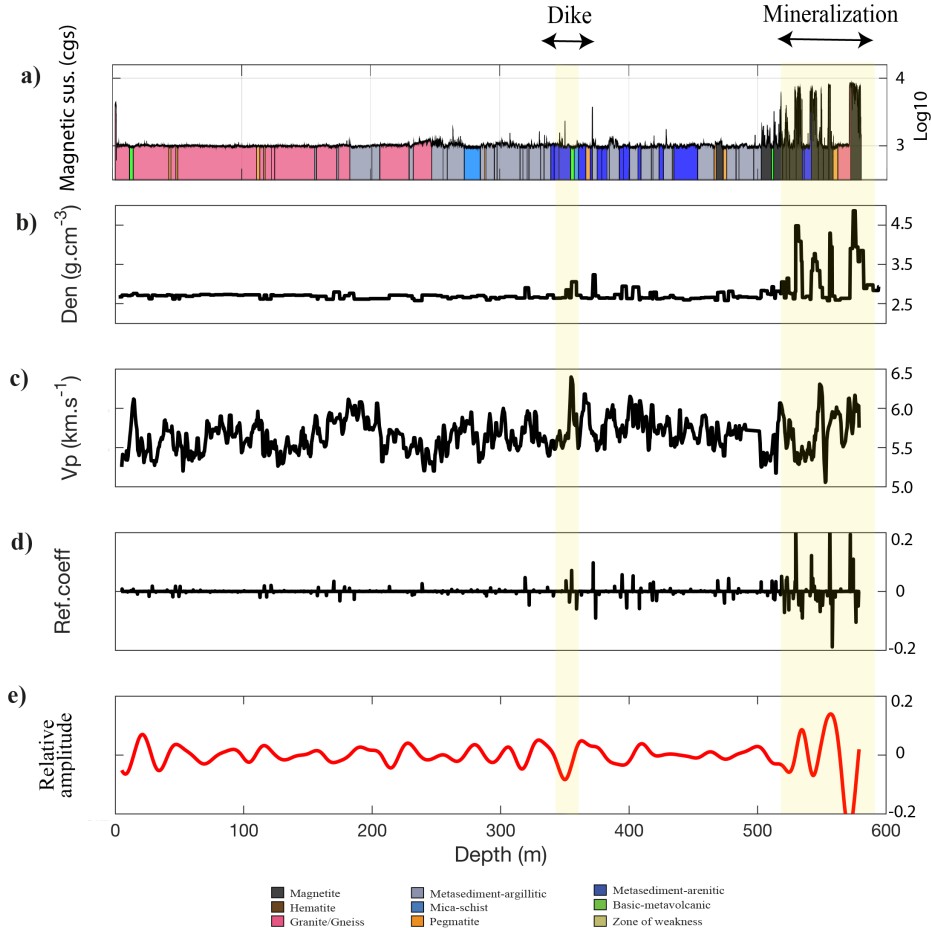

**Figure 8.** Logging data from BB14004. (a) Natural gamma and rock types, (b) Density, (c) P-wave velocity values and (d) Calculated reflection coefficients. (e) The synthetic trace using a 70-Hz Ricker wavelet. The iron-oxide deposits (530-570 m) produced strong seismic signal in the synthetic trace.

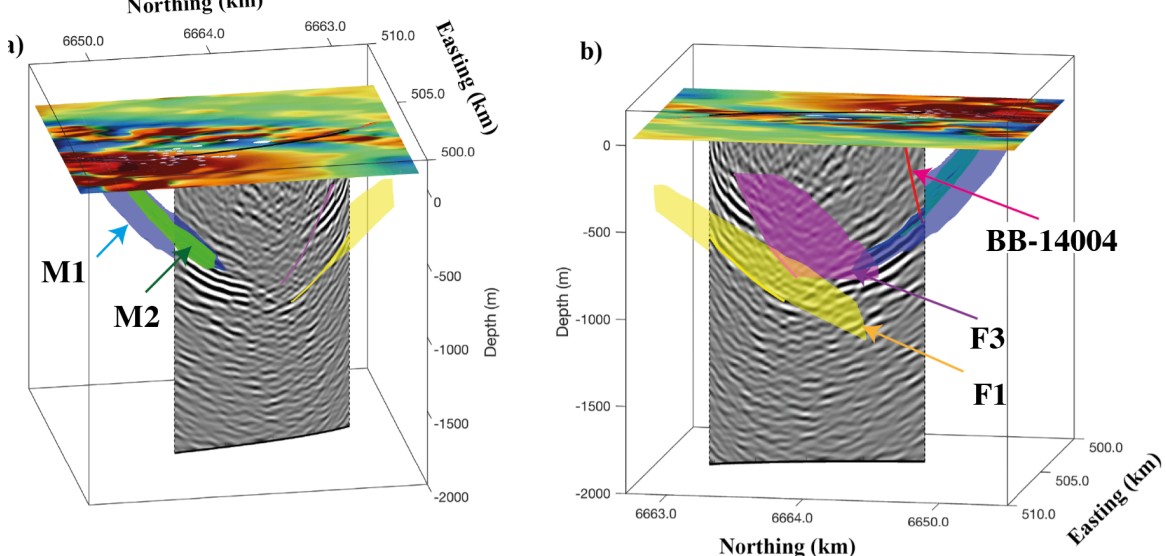

Figure 9. 3D view integrating the RTM section with other geophysical data. (a) M1 and M2 are 3D ore block models intersecting the strong reflectors in the seismic section. These imaged strong reflectors also correlate well with the high magnetic anomaly seen on the northern part of the profile in the overlaid magnetic map. (b) Two inferred fault planes (F1 and F3) as mapped by the recent 3D seismic data in the area and borehole BB-14004 (solid red line) intersecting the deposits. A small discrepancy between the imaged reflectors and the inferred fault planes (F1 and F3) might be due to the out-of-the-plane nature of these features producing a biased dip in the 2D RTM plane.

We interpreted the seismic image in the contexts of the four different datasets as above-mentioned. For a better illustration,
we also show the integrated datasets along the 2D seismic section (Figure 10). First, we extracted the magnetic anomaly data
along the seismic acquisition line (Figure 10a). Based on the high values of the magnetic anomaly data in the northern east
part along the profile, the position of the deposits is indicated well. From 1000 m to 1500 m distance along the profile, the
magnetic anomaly bulges up again, which we related to a shallow reflector (L2 in Figure 10b) in the seismic section. Second,
we plotted the intersections between the two mineralized sheets and the RTM image (M1 and M2 in Figure 10b). The two
intersection lines match well with the negative peaks in the seismic section. Based on the continuity of the reflectors at the
deposit depths, it is highly reasonable that the mineralised sheets extend down to approximately 1000 m than the ore block
model. We also marked M3 (Figure 10b) as a potential mineralization below M1 and M2.  Third, we marked out the
intersections between the two inferred fault planes and the RTM image (F1 and F3 in Figure 10b). The intersection due to the
deeper fault matches well with the reflector, which is cross-cutting the deposit reflectors. However, the intersection due to the
shallow fault matches well only at the deeper part around the deposits. Fourth, we projected the 1D synthetic seismic trace
from BB-14004 on the 2D image (Figure 10b). The large amplitudes in the synthetic seismic trace at a depth interval of 500-
600 m matches well with the reflectors in the image. Additionally, the reflector imaged above the deposits also matches well
with weak amplitude in the synthetic seismic trace at a depth interval of approximately 350-370 m (L1 in Figure 10b). We
attributed this reflector to a dike (Figure 8) based on the borehole core logging information. There are other reflectors clearly

1  shown up in the RTM image. However, we did not interpret them due to the lack of other geological and geophysical data at

2  their locations.

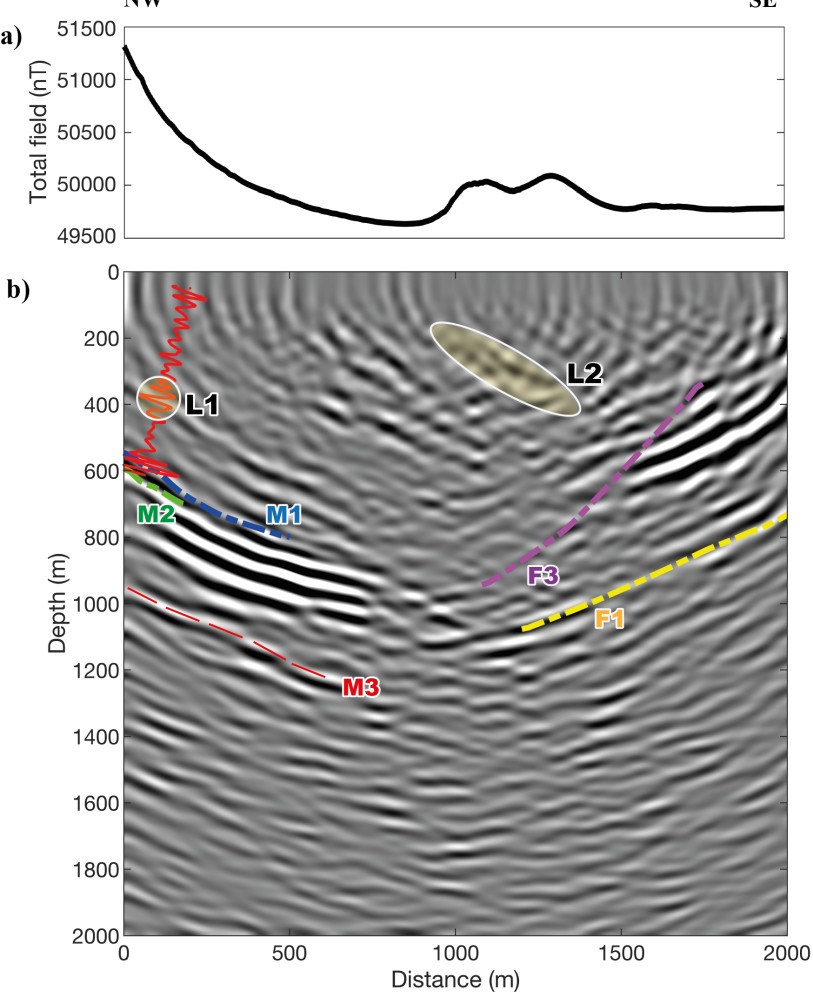

**Figure 10.** (a) Total-field magnetic anomaly along the seismic section. (b) The intersections of two deposit surfaces (M1 & M2) and the intersection of the two fault planes (F1 & F2) are plotted. The synthetic trace is plotted along the well trajectory. R1 is interpreted to be from a lithological contact (see Figure 8). R2 may be a weak iron-oxide mineralization as it appears in a region with slightly higher magnetic properties than the neighbouring areas.

A specific comparison was made between the produced RTM image (Figure 11a) and the image (Figure 11b) obtained by
running a poststack Kirchhoff migration using two datasets acquired in 2015 and 2016 in the same area (Markovic et al., 2020).
Though the main features of the mineralization are similar in the two images, the RTM section imaged one reflector (L1 in
Figure 11a) clearly above the mineralization and indicated well the two possible crosscutting features (F1 and F3 in Figure
11a). Even the RTM image was obtained by using dataset acquired in 2016 only, it showed more details because of the RTM
imaging algorithm.
We also made comparisons of the RTM images produced from different datasets which were pre-processed using different
methods in the pre-processing flow. In this way, we demonstrated the influences of different pre-processing methods on the
final RTM image results. To show the influence of the offset regularization (Step 4 in Figure 2), we relocated the seismic
traces in the original dataset to be at the points of 5-meter grids along the 1D curve without trace interpolations. The relocation
of any seismic trace is simply by moving the trace to the nearest point of the 5-meter grids relative to its true receiver location.
Keeping other pre-processing steps unchanged, the resultant RTM image (Figure 11c) without offset regularization was
providing similar information in the area with 5 m receiver spacing designed, compared to the RTM image (Figure 11a)
obtained with offset regularization. However, the offset-regularized data improved fault imaging (ellipse in Figure 11a) in the
10 m receiver spacing designed area since the trace density were doubled in this area. To show the effects of different methods
for surface-wave suppression (Step 5 in Figure 2), we tested the widely used median filter to suppress the surface waves
without changing other pre-processing steps. We found that the RTM section (Figure 11d) obtained by using the data pre-
processed with median filter did not present the obvious crosscutting feature F1 which were shown in the RTM section (Figure
11a) with the curvelet filter in the pre-processing.

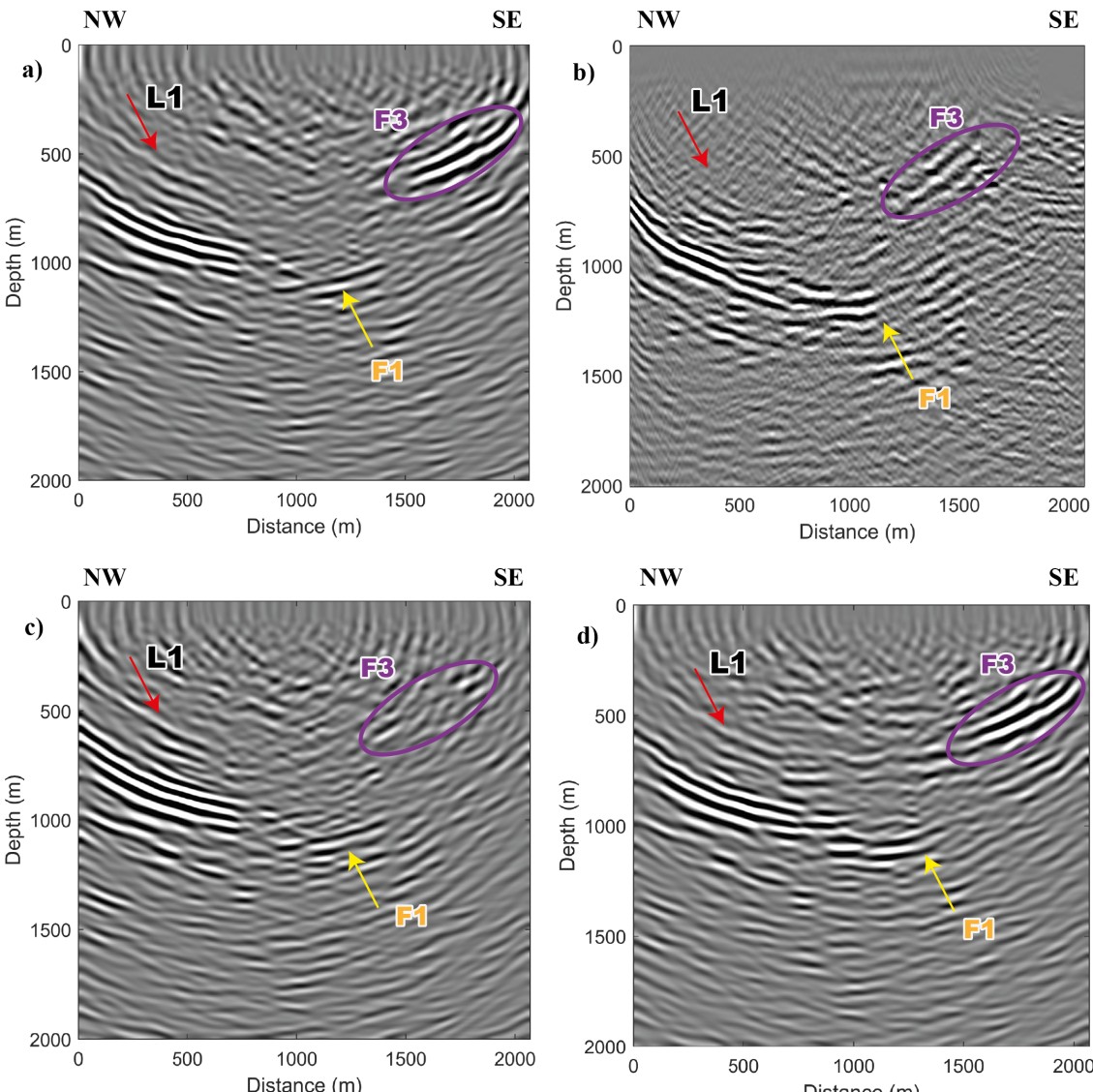

2 **Figure 11.** Comparisons of different image results. (a) The RTM image (same as Figure 10). (b) Image produced by a poststack Kirchhoff
3 migration in Markovic et al. (2020). (c) RTM image obtained from data without offset regularization in the pre-processing, step 4. (d)
4 RTM image obtained from data with median filter applied in the pre-processing, step5.

## 5 Discussion

In the current 2D case, when considering the sheet-like targets having a dip at certain depth, one needs to set up the orientation and the length of the seismic line accordingly. The proper orientation (i.e., perpendicular to the strike of the targeted dipping layer) of the profile ensures obtaining a nearly true dip of the target in the image. The proper length of the profile allows receiving the reflected signal from the target at depth.

In our study, the six-step pre-processing workflow was essential in preconditioning the data and hence obtaining a good RTM seismic section. This pre-processing workflow is recommended for future studies of RTM on hardrock seismic data, though different methods could be used to preprocess the data as well The migration velocity model is another main factor that influences the accuracy of the resultant RTM image. If the data itself lacks reflection events to be used for a good velocity analysis across the whole velocity model, one needs to be careful with the interpretation of the reflectors in the final image. With the CIG analysis (e.g., Schleicher et al., 2008), it is possible to refine the velocity model in the future studies. Based on the flatness of those events in CIG images, we argue that the current velocity model works well for imaging the mineral deposits at the site, though the migration velocity needs to be updated for better imaging other subsurface structures. An accurate 2D velocity model using VSP surveys could likely improve the RTM imaging results. However, the current study already supports the RTM imaging methods to be attempted for hardrock seismic datasets and for mineral exploration purposes.

A remaining topic that one should also consider is that if there is strong AVO/AVA (amplitude-versus-offset/angle) effect (Castagna and Backus, 1993; Hilterman et al., 2000) in the CIG images. Using Zoeppritz equations (Zoeppritz, 1919; Sheriff and Geldart, 1995), we calculated the AVA (amplitude-versus-angle) effect using a simple isotropic two-layer model (Figure 12a), which simulates physical properties of crystalline rocks but rather in a layered case. The amplitudes of P- and S-waves reflected from the horizontal contact between the granite and the iron-oxide mineralization show a strong variation (Figure 12b) versus the incident angles of the plane waves (P). Relevant to this specific legacy dataset, the dipping angle and the thickness of the mineralized sheets need further to be accounted for when studying their AVO effects. Since the CIGs produced from the RTM partial images are ideal for studying the AVO effect relative to the CIG locations (Yan and Xie, 2012), future studies should exploit this potential for better scrutinizing and extracting physical property information from seismic data. Additionally, crystalline environment with high degree of metamorphism may even show strong anisotropy to cause anisotropic AVO response (Asaka, 2018). Utilizing the isotropic/anisotropic AVO analysis as a supplementary tool to characterise various rocks and mineral deposits from the seismic data is ideal in mineral exploration (Harrison and Urosevic, 2012).

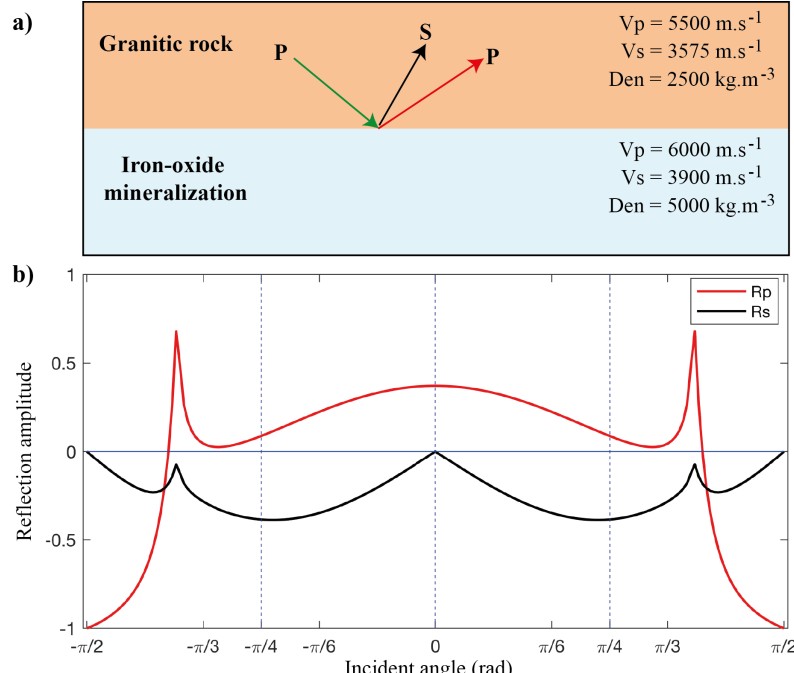

Figure 12. (a) A two-layer model with rock properties of granitic rock in the top layer and that of iron-oxide mineralization in the bottom layer. (b) The amplitude versus incident angle (AVA) response of the reflected P- and S-waves. The incident plane wave is P-wave with amplitude 1.

## 6 Conclusions

We have studied the application of RTM imaging method on a hardrock seismic dataset acquired for deep-targeting iron-oxide deposits in the Ludvika mining area of central Sweden. Using a 6-step pre-processing workflow, we suppressed the unwanted noise and improved the signal-to-noise ratio. Consequently, the reflected events from the deposits and other geological features were remarkably strengthened. The resultant RTM image shows clearly several reflectors, which are consistent when compared with four other independent datasets. From the known deposit model constrained from existing boreholes, two sets of strong seismic reflectors match well with the two iron-oxide mineralised bearing horizons. Two oppositely dipping reflectors, interpreted to be from fault planes, intersect the two strong reflectors from the mineralization implying possibly a geological control on the extension or termination of these deposits at depth.

Integrating the seismic image with the high-resolution magnetic anomaly data, a weak zone of iron-oxide mineralization can be interpreted at shallow depth. Using P-wave sonic data, density and core logging data, we identified one continuous reflector as the dike formation. AVO effect was also studied using a simple two-layer model since we speculated a possible AVO response in the CIGs. There may be opportunities for detailed AVO studies of dense metallic deposits in either theoretical modelling or real field applications. In summary and exemplified with the Ludvika legacy seismic dataset, we demonstrate the

usefulness of advanced imaging methods such as RTM for deep targeting and imaging mineral deposits and their host rock structures.

*Code Availability.* This work has benefited from the open-source software 'CREWES'.

*Data Availability*. Original data underlying the material presented are available by contacting the corresponding author. However, as the dataset is the subject of other PhD studies, there is a period of 3 years embargo on their availability.

*Author contributions*. AM contributed to the data acquisition. YD worked on the data processing, and wrote the paper, with contributions from AM.

*Competing interests.* The authors declare that they have no conflict of interest.

*Special issue statement.* This article is part of the special issue "State of the art in mineral exploration". It is a result of the EGU General Assembly 2020, 3–8 May 2020.

*Financial support.* This research has been supported by the Euro- pean Commission, Executive Agency for Small and Medium-sized Enterprises (Smart Exploration grant no. 775971).

*Acknowledgements.* This work was performed within the Smart Exploration project. The Smart Exploration was funded by European Union's Horizon 2020 research and innovation program with the grant agreement no. 775971. We thank Nordic Iron Ore (NIO) AB for their collaboration in this study. Georgiana Maries is thanked for providing the density and sonic logs. We thank two anonymous reviewers and the editor for providing their helpful comments to improve the original version of the manuscript.

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
