# Peer review of "Reverse time migration (RTM) imaging of iron-oxide deposits in the"

_Solid Earth, 2020_

## Referee Comment (RC1) · Anonymous Referee #1 · 25 Nov 2020

The review for "Reverse time migration (RTM) imaging of iron-oxide deposits in the Ludvika mining area, Sweden"

The authors presented an interesting case study with application of Reverse Time Migration which has been rarely practiced in the hard rock environment. The study provides some clues for those would like to test/improve the method for mineral exploration.

In Figure 2, the authors show their processing flow. Why they did not applied deconvolution to remove seismic source effect. Normally, deconvolution is a crucial step in hard rock environment. The authors need to address if they applied it and if not then

why? Is it crucial to preserve the seismic source signature in the shot gathers when RTM is applied?

In Step 4 offset regularization: How did you apply it. Did you need to pick specific reflection in a shot gather and try to make it up in which to fill the area that the reflection is missing/improve the coherency of the reflection? Or, you applied the linear interpolation filter equally to all shot gather? Please explain this step in more details. Also, can you discuss the pros/cons of the offset regularization method in hard rock environment? Step 5: what is advantage of curvelet filtering method to remove surface waves? Is it working better than median filter? Did you test both filters (i.e., median versus curvelet filter)?

What is the best procedure to improve the migration velocity model shown in Figure 4? Please explain in more details.

In this case study the straight part of the survey is considered. How do you deal with a crooked survey? Do you think the RTM method is applicable? Please provide more insight about crooked surveys.

---

## Referee Comment (RC2) · Anonymous Referee #2 · 29 Nov 2020

This article presents Reverse-time migration applied to Ludvika mining area seismic reflection data. The article contains some interesting view points but it remains in a level of "least publishable unit": most interesting ideas such as AVO/AVA are just briefly mentioned and left for future studies, also more detailed testing of influence of different processing and migration velocity field.

In the current form, this article provides very little new interesting insight to the reader. If the ideas presented were developed and discussed in more detailed and scientific manner, this could make a really interesting paper. From the view point of the data interpretation, it remains unclear if this study brought any new insight into the geology

compared to previous work published about the same dataset.

Attached file contains more detailed comments and suggestions for authors to improve their work.

Please also note the supplement to this comment:
https://se.copernicus.org/preprints/se-2020-181/se-2020-181-RC2-supplement.pdf

[Figure]

**Supplement:**

[revised manuscript text omitted]

---

## Author Comment (AC1) · 18 Jan 2021

Dear reviewer,

We appreciate your positive feedback. We address your comments point-by-point below. Our revised manuscript will cover these aspects.

The review for "Reverse time migration (RTM) imaging of iron-oxide deposits in the Ludvika mining area, Sweden"

The authors presented an interesting case study with application of Reverse Time Migration which has been rarely practiced in the hard rock environment. The study

provides some clues for those would like to test/improve the method for mineral exploration.

In Figure 2, the authors show their processing flow. Why they did not apply deconvolution to remove seismic source effect. Normally, deconvolution is a crucial step in hard rock environment. The authors need to address if they applied it and if not then why? Is it crucial to preserve the seismic source signature in the shot gathers when RTM is applied?

[Reply: We agree that deconvolution is a crucial step when performing migration algorithms. Though we didn't apply any deconvolution methods to the data in the pre-processing step, the imaging condition in RTM is a deconvolution imaging condition (Claerbout, 1971), which removes the source wavelet approximately, and potentially improves illumination compensation. We add one sentence to make this clear in section 3.3 RTM imaging as below.

'Such a deconvolution imaging condition (Claerbout, 1971) removes the source wavelet approximately and hence improves the resolution of the image.'

Ref: Claerbout, J. F., 1971. Toward a unified theory of reflector mapping: Geophysics, 36, no. 3, 467–481]

In Step 4 offset regularization: How did you apply it. Did you need to pick specific reflection in a shot gather and try to make it up in which to fill the area that the reflection is missing/improve the coherency of the reflection? Or, you applied the linear interpolation filter equally to all shot gather? (We applied it to all the shot gathers equally) Please explain this step in more details. Also, can you discuss the pros/cons of the offset regularization method in hard rock environment?

[Reply: To show how the offset regularization is done, we add one more figure (Figure 4) in our revision. We described regularization in detail in section 3.2 Data Preprocessing as below.

'The sources and receivers were projected onto a smooth curved profile, which was defined by a third-degree polynomial (Figure 4). Based on the projected receiver locations, we regularized the receiver spacing to a constant interval of 5 m along the 1D-curved profile. The seismic traces at those regularized receiver locations were obtained by a cubic interpolation using the traces at the projected receiver locations. Similarly, we regularized the source locations along the curved profile and obtained the seismic traces at every 5 m interval by a cubic interpolation, which was performed in the common receiver domain.'

The pros/cons of the offset regularization for running RTM is discussed in section 3.3 RTM imaging as below.

'In this study, we found that the regularized data in the area where receiver spacing was roughly 5 m did not improve the final image, however, the regularized data in the area where receiver spacing was approximately 10 m (southern portion of the seismic profile) provided improved images because the trace density was doubled in this area.']

Step 5: what is advantage of curvelet filtering method to remove surface waves? Is it working better than median filter? Did you test both filters (i.e., median versus curvelet filter)?

[Reply: In our study, curvelet filtering is used to extract the surface waves in local regions. It is advantageous to only apply filters to a local region where the noise is dominant and leave the data in other regions unaffected.

We tested the median filter as suggested. Using the median filter, the median values were extracted by running a sliding window along several specific slopes and then subtracted from the data. We found that the median filter tends to harm the reflected signals since it is applied globally to the entire data.

We add one figure (Figure 7) showing the RTM image using the median-filtered data while keeping other pre-processing steps unchanged. In the image, we see that the

crosscutting feature is not as obvious as in the RTM image obtained from the curvelet-filtered data.

To remind potential readers that our chosen methods for pre-processing are not unique, we add one sentence in section 3.2 Data Pre-processing as below.

'Though we chose the specific processing methods in the pre-processing steps to prepare our data for RTM, it is possible to use different methods for pre-processing when done carefully.']

What is the best procedure to improve the migration velocity model shown in Figure 4? Please explain in more details.

[Reply: There are many ways (e.g., full waveform inversion, reflection/refraction tomography, etc.) to perform a velocity model building. But the velocity model building is limited by the seismic data quality and offsets. As for our dataset, the data is rather noisy for any of these approaches and first arrivals do not sample very deep due to a maximum offset of only 2.5 km. Besides the semblance velocity analysis (which is not so useful for hardrock datasets), we actually tested 10 constant velocity models from 5100 m/s to 6000 m/s with an interval of 100 m/s by running Kirchhoff migrations. Then we found velocity 6000 m/s gave us the best result of the reflections from the mineralization area, which flattens the reflections most in the common image gathers. Based on this constant velocity, we modified the shallow area of the model to make the reflection events in the common image gathers as flat as possible when running RTM. Readers to be reminded that the dataset is already corrected for refraction statics therefore theoretically near-surface low velocities are compensated for at the step of velocity model building.]

In this case study, the straight part of the survey is considered. How do you deal with a crooked survey? Do you think the RTM method is applicable? Please provide more insight about crooked surveys.

[Reply: It is possible to apply RTM along a crooked line if we parameterize the locations of sources and receivers by offsets only. In such a situation, 2D RTM can be run along a crooked line. However, we need to be aware that the strong crooked line may create artifacts in the image, which are difficult to be identified and separated from the true imaged reflectors.

A safer way is to run RTM on several approximately straight segments of the acquisition line independently, then a final image is obtained by merging the images from those segments.]

————————————————————

---

## Author Comment (AC2) · 18 Jan 2021

Dear reviewer,

Thank you for the detailed and insightful comments. We have copied your comments with Page# and Line# showing from the supplement file and address them point-by-point below.

This article presents Reverse-time migration applied to Ludvika mining area seismic reflection data. The article contains some interesting view points but it remains in a level of "least publishable unit": most interesting ideas such as AVO/AVA are just

briefly mentioned and left for future studies, also more detailed testing of influence of different processing and migration velocity field.

[Reply: The AVO effects were speculated when we produced the common image gathers to form the final RTM image. We would like to keep our focus on the RTM imaging for mineral exploration and give details on how we performed the processing flow and the velocity model building, while expanding the AVO effects in the manuscript tends to be out of the scope. We did not want to ignore this in the interpretation section and agree that one needs to explore this further and in details since it would open new possibilities and improve interpretations for such a study.]

In the current form, this article provides very little new interesting insight to the reader. If the ideas presented were developed and discussed in more detailed and scientific manner, this could make a really interesting paper. From the view point of the data interpretation, it remains unclear if this study brought any new insight into the geology compared to previous work published about the same dataset.

[Reply: Our goal in this manuscript is to show an application of RTM imaging in mineral exploration, compared with Kirchhoff migration results which are done by previous studies (e.g., Bräunig et al., 2020 in Geophysical Prospecting). Though interpretations of geology were given, which were based on previous imaging results, it is still helpful to see if RTM as a two-way wave equation imaging technique can show more subtle details in the resultant image. As in our RTM image, the rock contact above the mineralization is clearly imaged while it was not imaged as successfully in the previous imaging workflows. Also, the cross-cutting feature is clearer when compared to the previous results.]

Attached file contains more detailed comments and suggestions for authors to improve their work.

Please also note the supplement to this comment: https://se.copernicus.org/preprints/se-2020-181/se-2020-181-RC2-supplement.pdf

[Reply: Thank you for the detailed comments and suggestions, we replied to them carefully as below.]

Page 1 Line 8: 'qualified choice' and 'resolution power at depth'

[Reply: We revised the sentence as below.

'To discover or delineate mineral deposits and other geological features such as faults and lithological boundaries in their host rocks, seismic methods are preferred to be used for imaging the targets at depth.']

Page 1 Line 8: what other goals is there?

[Reply: Based on different goals, the seismic data can be used for velocity model building, seismic attributes analysis, reflector imaging, etc. In our context, we meant that the main goal of our study is to image the reflectors, though it can be related to other goals during the processing and imaging. We revised the sentence as below.

'One major goal for seismic methods is to produce a reliable image of the reflectors in the subsurface.']

Page 1 Line 11: what makes it "legacy" data set?

[Reply: We deleted the word 'legacy' in the context.]

Page 1 Line 13: fidelity

[Reply: Fidelity specifies how accurately the targeted features have been mapped at the correct locations (Zhou et al., 2018). Though the resolution can be limited due to the source itself and background velocity, the fidelity can be improved since RTM which obeys the two-way wave propagation maps the data more to their true locations.

Ref: Zhou, H., Hu, H., Zou, Z., Wo, Y. and Youn, O.: Reverse time migration: A prospect of seismic imaging methodology, Earth-Sci. Rev., 179, 207–227, doi:10.1016/j.earscirev.2018.02.008, 2018.]

Page 1 Line 15: aren't these important factors for success of every survey & research?

[Reply: Yes, they are. We had no intension to mislead readers by stating that they are only important for RTM. However, we revised the sentence below.

'In order to obtain a reliable RTM image, we performed a detailed data pre-processing flow to deal with ambient and source-generated noise, near-surface effects and irregular receiver spacing and source spacing, which downgrade the final image if are ignored.']

Page 1 Line 17: what kind of detail exactly? At exact depth, do you get values of thickness, length of deposit, shape of deposit?

[Reply: The deposit is not a single and thick layer, but a composite of several layers of mineralization as shown in the logging data in our revised figure. Detection of the thickness of the deposits is difficult from the seismic image in which the main wavelength is around 100 m. Combined with the logging data, we did infer the vertical extent of the deposits based on the continuity of the imaged reflectors.]

Page 1 Line 17: improved compared to what?

[Reply: The sentence is revised as below.

'It also provides a much-improved image of the lithological contacts and crosscutting features relative to the mineralized sheets when compared to the images produced by Kirchhoff migrations in the previous studies.']

Page 1 Line 18: 'Some of the imaged' which exactly?

[Reply: The sentence was revised as below.

'Two of the imaged crosscutting features are considered to be crucial when interpreting large-scale geological structures of the site and the likely.'

We did discuss these two crosscutting features in section 4 RTM results and their

interpretations.]

Page1 Line 20: this is very weak message for a scientific paper: method to have potential to be used...

[Reply: The sentence is revised as below.

'Exemplified with a field data acquired in a hardrock environment, we demonstrate the usefulness of the RTM imaging workflow for deep targeting mineral deposits.']

Page 1 Line 23: they have been used, not only becoming promising...

[Reply: The sentence is revised as below.

'Seismic methods are favourable for deep targeting metallic deposits, because of their ability to image targets and retain a good resolution at great depth (+500 m).']

Page 1 Line 24-25: how is this comparison? EM methods can also use manmade source that causes electrical waves...

[Reply: We revise the sentence as below.

'Compared to other geophysical methods (e.g., gravity, magnetic and/or electromagnetic), the seismic methods investigate the properties (i.e., impedance) of the subsurface in a wave-equation-based way in processing and imaging. Generally speaking, due to the less attenuation effects of the seismic waves as a function of depth compared to EM methods, they tend to hold better resolution at depth than EM methods although they do have different sensitivity to different properties.']

Page 1 Line 27: 'qualified' what does this mean?

[Reply: Deleted.]

Page 1 Line 28: what conditions? what means properly desingned? When conditions are not met and what is poor design of survey?

[Reply: The sentence is revised as below.

'Among various geophysical methods, seismic methods can provide an image of the targets with high resolution at depth, when the survey is designed to record the signal reflected from the targets directly below the survey area/line.']

Page 1 Line 28: what is proper source? dynamite, vibroseis, weightdrop, ...??? is one of these more proper than other and why?

[Reply: We deleted the word 'proper'. We did not intend to compare different sources in this manuscript.]

Page 2 Line 29: how to ensure sufficient energy?

[Reply: The more weight of the impact source or the more vibrating energy produced from the vibrator, the more seismic energy into the subsurface. In the field, there will be a few field tests to be performed to check the data quality to make sure that the source has sufficient energy propagated down and reflected back from the target depth before the real acquisition starts. This procedure usually considers target depth and size as well as how source-generated source becomes dominant compared to the wanted reflected signal.]

Page 2 Line 30-31: what is reasonable in this context? give example of frequency range...

[Reply: Targets of different sizes and depths require different minimum frequency range to image them clearly. In our case, the frequency range is from 35-180 Hz. The dominant frequency in the data is around 60 Hz. With a background velocity of ~6000 m/s, consistent with the logging data, the dominant wavelength is approximately 100 m. We will add this information in the description of the dataset in the section 3.3 RTM imaging.]

Page 2 Line 33: have 'well' been established

[Reply: Corrected. have been well established]

Page 2 Line 36-37: how is this related to the goal of delinating deep targets? Of course, they are costlier to exploit, but still there is urge to do it.

[Reply: We deleted the word 'exploiting'.]

Page 2 Line 39: ??? if they are not continuous, one obviously cannot image them seismically continuous

[Reply: We agree with this comment. We revised the sentence as below.

"Second, the strong scattered waves due to abundant small heterogeneities and complex structures in the near surface (e.g., glacial tills in the case of Blötberget dataset) (Cheraghi et al., 2013; Bellefleur et al., 2018; Braİ́Lunig et al., 2020) will containment the reflections from targets of mineralization and cause difficulties to image the reflectors."]

Page 2 Line 41: main factor is, that for deep exploration there hardly is a better method than seismic

[Reply: We agree with the reviewer on this fact. But this fact will not ensure that seismic methods can be welcomed widely in the industry and academia.

Page 2 Line 46: 'with efforts'

[Reply: Corrected. with effort]

Page 2 Line 50: 'in a legacy dataset'

[Reply: Corrected. to a field dataset]

Page 2 Line 51: 'from' the central Sweden

[Reply: Corrected. in the central Sweden]

Page 2 Line 53: two-way-wave

[Reply: Corrected. two-way wave]

Page 3 Line 66: 'improved' resources

[Reply: Changed. more available resources]

Page 3 Line 69: 'kg.m-3'

[Reply: Corrected. 'kg m-3']

Page 4 Line 83 'receivers' spacings'

[Reply: Corrected. receiver spacing]

Page 4 Line 84 use approximately in previous sentence.

[Reply: The sentence was deleted and the previous sentence was revised as below.

'As for the receiver spacing, the cabled recorders were deployed at every 5 m approximately while the wireless recorders at every 10 m approximately.']

Page 4 Line 90: Markovic-Juhlin

[Reply: Corrected. Markovic Juhlin]

Page 5 Line 109-110: how much did the original spacing differ from 5 m? some cm? if you did not do this, how would image look like?

[Reply: We will add one figure to show the irregularities of receiver spacing]

Page 5 Line 112: 'with cares'

[Reply: Corrected. with care]

Page 7 Line 140: why only relatively good, why not best possible?

[Reply: It was the best possible using this dataset. We will rephrase it in the context]

Page 8 155: why do you think so? why is it "promising" and what does it promise? show results from other studies...

[Reply: To show the promising aspects of RTM results, the RTM image will be compared to one previous section (Markovic et al., 2020), which is obtained by a poststack Kirchhoff migration (PSKM). Please notice that two datasets acquired in 2016 and 2015 were both used to produce the PSKM image, while we only used the data acquired in 2016 in the current RTM work. We will add more discussions on this comparison in section 3.3 RTM imaging with one additional figure.]

Page 9 183-184: Reference? Where does this data originate from?

[Reply: Thanks for pointing this out. The magnetic data were provided by the Geological Survey of Sweden. This is information was added in the caption of Figure1.]

Page 10 Figure 7: what are the rocks?

[Reply: We will add the rock type in Figure 'Logging data from BB14004'.]

Page 12 Figure 9 b): what are all reflections below the known mineralization?

[Reply: The mineralization is not only two layers, it contains a few cycles of mineralized sheets.]

Page 12 Figure 9 b): what are these strong reflectors?

[Reply: They were interpreted as the fault plane, which is F1. The mismatch between the strong reflector in 2D RTM image and the F1 picked from a 3D image volume is suspected due to that the fault plane is out of the acquisition plane.]

Page 12 Line 220-222: this paper did not discuss design of the seismic survey and showed one way of processing the data. I do not see much of discussion point in this.

[Reply: We removed this part as suggested.]

Page 12 Line 224: just like any other surveys. Remove.

[Reply: Removed.]

Page 12-13 Line 225-227: please then tell, what is proper orientation and what not.

surely this acquisition line, as so many others, are also restricted by existing roads and not actually designed in optimal way regarding geology...

[Reply: we revised the sentence as below.

'The proper orientation (i.e., perpendicular to the strike of the targeted dipping layer) of the profile ensures obtaining a nearly true dip of the target in the image.']

Page 13 Line 228: are you sure you could not have done things differently and still get good enough RMT image?

[Reply: Please notice that we stated the task of every pre-processing step in Figure 1, but we did not relate the task to any specific method in our flowchart. Though we chose specific methods to pre-process our data, it is still possible to achieve similar RTM results through other different pre-processing methods. To test different pre-processing methods is not our focus, but rather to demonstrate the usefulness of the RTM in mineral exploration using seismic methods. We did a comparison between our RTM image and one Kirchhoff image from the previous studies (Markovic et al., 2020) to show that the RTM imaging algorithm did produce an improved section even with less data used.]

Page 13 Line 230: did you try different models? how did the image change?

[Reply: We actually tried 10 constant velocity models before choosing the most suitable one. Using this most suitable constant velocity model, we did semblance velocity analysis to make sure the events are flattened in the CMP gathers where this was possible to be noted by an eye. Then, we updated the velocity model slowly to make the noticeable reflection events to be as flat as it can be in the CIG gathers when running the RTM. The velocity we used was obtained with many trials by checking the final RTM image and its CIGs iteratively.]

Page 13 Line 233-235: should be done already in this study and update it for the review.

[Reply: Similar to the above-raised comments: We did many trials and updates of

the model to obtain the best one using only this dataset. If there are other datasets available to be used, it may be possible to update the velocity model. Without any other seismic datasets used, we want to convince the potential readers that we can build a good velocity model from the dataset alone for a reliable RTM image.]

Page 13 Line 236: 'even hardrock seismic'

[Reply: Corrected. However, the current study already supports the RTM imaging methods to be attempted for hardrock seismic datasets and for mineral exploration purposes.]

Page 13 Line 245: why future studies and not this particular study?

[Reply: Our focus of this manuscript is to demonstrate the usage of RTM in mineral exploration. We did the simple numerical simulation of AVO effects in the discussion because we speculated the amplitude variance along the offsets in the CIG gathers. A detailed AVO study due to metallic deposits in a field dataset will be a total independent work, which is out of our scope in this manuscript. We know it has a great value and contributions in the topic!]

Page 15 Line 269: 'we illustrate the potential of'

[Reply: Changed. 'We demonstrate the usefulness of']

---

## Referee Report (RR1)

[revised manuscript text omitted]

I am not sure here. why this is here. You did not do this work.

[Figure]

**Figure 12.** (a) A two-layer model with rock properties of granitic rock in the top layer and that of iron-oxide mineralization in the bottom
layer. (b) The amplitude versus incident angle (AVA) response of the reflected P- and S-waves. The incident plane wave is P-wave with
amplitude 1.

**6 Conclusions**

We have studied the application of  RTM imaging method on a hardrock seismic dataset acquired for deep-targeting iron-oxide deposits in the Ludvika mining area of central Sweden. Using a 6-step pre-processing workflow, we suppressed the unwanted noise and improved the signal-to-noise ratio. Consequently, the reflected events from the deposits and other geological features were remarkably strengthened. The resultant RTM image shows  several reflectors, which are consistent when compared with four other independent datasets. From the known deposit model constrained from existing boreholes, two sets of strong seismic reflectors match well with the two iron-oxide mineralised bearing horizons. Two oppositely dipping reflectors, interpreted to be from fault planes, intersect the two strong reflectors from the mineralization implying possibly a geological control on the extension or termination of these deposits at depth.

Integrating the seismic image with the high-resolution magnetic anomaly data, a weak zone of iron-oxide mineralization can be interpreted at shallow depth. Using P-wave sonic data, density and core logging data, we identified one continuous reflector as the dike formation. AVO effect was also studied using a simple two-layer model since we speculated a possible AVO

response in the CIGs. There may be opportunities for detailed AVO studies of dense metallic deposits in either theoretical modelling or real field applications. In summary and exemplified with the Ludvika legacy seismic dataset, we demonstrate the

*last paragraph is a mess. It has 3 separate ideas. Perhaps just combine with 1st paragraph as a list of conclusions.*

[revised manuscript text omitted]

---

## Author Response (AR2)

Dear Editor Ramon Carbonell,

We appreciate greatly your time and efforts on handling with our manuscript. We are thankful for the two reviewers helping us to improve the manuscript in language and technical issues.

We have carefully revised manuscript based on the reviewers' questions and comments point by point.

We did all corrections in language-related issues. For the technical issues,

1. We revised Step 1 in 3.2 Data Pre-processing to remove the contradictive statements (Please check Page 5 Line 9—10).
2. We explained the small moveout in the reflection shown in Figure 3 (Please check Page 6 Line 10—12).
3. For the AVO effects caused by sheet-like mineralized deposits, we would like to keep it in Discussion because we want to remind the potential readers that the widely used AVO study for exploring oil and gas might help to discover mineral deposits in mineral exploration.

Kind Regards,
Yinshuai